# Buffered EGFR signaling regulated by *spitz*-to-*argos* expression ratio is a critical factor for patterning the *Drosophila* eye

**Nikhita Pasnuri**[1], **Manish Jaiswal**[1], **Krishanu Ray**[2,3], **Aprotim Mazumder**[1]*

**1** Tata Institute of Fundamental Research Hyderabad, Gopanpally, Serlingampally Mandal, Hyderabad, Telangana, India, **2** Department of Biological Sciences, Tata Institute of Fundamental Research, Navy Nagar, Colaba, Mumbai, Maharashtra, India, **3** National Brain Research Centre, Manesar, Haryana, India

* aprotim@tifrh.res.in

**Data Availability Statement:** All relevant data are within the manuscript and its Supporting Information files.

## Abstract

The Epidermal Growth Factor Receptor (EGFR) signaling pathway plays a critical role in regulating tissue patterning. *Drosophila* EGFR signaling achieves specificity through multiple ligands and feedback loops to finetune signaling outcomes spatiotemporally. The principal *Drosophila* EGF ligand, cleaved Spitz, and the negative feedback regulator, Argos are diffusible and can act both in a cell autonomous and non-autonomous manner. The expression dose of Spitz and Argos early in photoreceptor cell fate determination has been shown to be critical in patterning the *Drosophila* eye, but the exact identity of the cells expressing these genes in the larval eye disc has been elusive. Using single molecule RNA Fluorescence in situ Hybridization (smFISH), we reveal an intriguing differential expression of *spitz* and *argos* mRNA in the *Drosophila* third instar eye imaginal disc indicative of directional non-autonomous EGFR signaling. By genetically tuning EGFR signaling, we show that rather than absolute levels of expression, the ratio of expression of *spitz*-to-*argos* to be a critical determinant of the final adult eye phenotype. Proximate effects on EGFR signaling in terms of cell cycle and differentiation markers are affected differently in the different perturbations. Proper ommatidial patterning is robust to thresholds around a tightly maintained wildtype *spitz*-to-*argos* ratio, and breaks down beyond. This provides a powerful instance of developmental buffering against gene expression fluctuations.

## Author summary

Sexual multicellular organisms start life as a single cell–the fertilized egg. One of the fundamental questions of Developmental Biology is to understand how cells proliferate and assume specific identities to faithfully reproduce the organismal tissue patterning. Cells communicate via signaling pathways to achieve complex patterning outcomes. Epidermal Growth Factor Receptor (EGFR) signaling is known to coordinate both cell division and fate choices in animals ranging from humans to the fruit-fly (*Drosophila melanogaster*). The fruit-fly eye with its strikingly patterned, hexagonally arranged units is a remarkable example of tissue patterning by EGFR signaling. In this paper, we investigate how

**Funding:** This project was supported by intramural funds at TIFR Hyderabad from the Department of Atomic Energy, Government of India (Project Identification No. RTI 4007 to AM). MJ is a Ramalingaswami fellow, Department of Biotechnology, Government of India, under project number BT/RLF/Re-entry/06/2016. The funders had no role in study design, data collection and analysis, decision to publish, or preparation of the manuscript.

**Competing interests:** The authors have declared that no competing interests exist.

expression levels of an activator (Spitz) and inhibitor (Argos) that tune the activation level of EGFR pathway regulate the patterning of the *Drosophila* eye. We find the activator-to-inhibitor ratio to be a critical factor in this process. The pattern is robust in a range around a tightly-controlled wildtype ratio. Beyond this biological range, both cell fate and cell division are affected producing different 'rough-eye' phenotypes. This is a striking example of how developmental patterning may be buffered against reasonable fluctuations in gene expression.

## Introduction

Receptor tyrosine kinases (RTKs) are key regulators of diverse cellular processes and development. Mutations or aberrant activation/inactivation of RTKs lead to different anomalies including cancers [1]. RTKs trigger a cytoplasmic signaling cascade involving Ras-MAPK pathway [2]. EGF receptor signaling is extensively studied for its role in cellular homeostasis and in relation to human diseases [2]. *Drosophila melanogaster* has one EGF receptor (DER) and four activating ligands [3]. The difference in expression of the ligands in a tight spatiotemporal pattern was suggested to bring about difference in EGF responses. Among *Drosophila* EGFR ligands, Gurken [4–6], Spitz [7–9] and Keren [10] are homologous to TGFα and Vein [11–13] is homologous to neuregulin. Spitz is the canonical EGF in the fly. Beyond different ligands, positive and negative feedback loops are known to tightly regulate EGFR signaling [14,15]. The downstream targets of EGFR include activators like Vein and Rhomboid [16,17] and inhibitors like Argos [18–20], Kekkon-1 [21,22] and Sprouty [23,24]. Of the feedback molecules, Sprouty, Rhomboid and Kekkon-1 act in a cell autonomous manner, while Argos is a diffusible factor like cleaved Spitz, and can potentially act both in a cell autonomous and non-autonomous manner. The competitive and stoichiometric sequestration of Spitz by Argos has been shown to be critical for patterning different tissues [18,25]. Spitz and Argos are thought to be short-range activator and long-range inhibitor respectively [26].

A strong instance of EGFR-mediated patterning occurs in the *Drosophila* compound eye with its periodic units called ommatidia. A single ommatidium comprises of eight neuronal photoreceptors (PR) accompanied by twelve non-neuronal cells [27–29]. The photoreceptor cell fate specification starts in the early 3rd instar larvae along an anteriorly progressing wave of Hedgehog in the eye disc leaving a morphogenetic furrow behind [30]. Posterior PR cells are older in developmental time than clusters just behind the morphogenetic furrow. EGFR signaling is a prerequisite for photoreceptor specification (except for founder photoreceptor, R8) and also cone and pigment cells until pupal stage [9,31].

Spitz binds to EGFR and activates the cytoplasmic Ras-MAPK pathway [32,33]. This cascade activates the transcriptional activator, PntP1 and degrades transcriptional repressor, Yan [34–36]. Downstream of the signaling pathway, *argos* is expressed and the protein product is secreted out of the cell. Argos binds Spitz in a 1:1 ratio by clamping to the EGF domain and restricts the amount of free ligand available for EGFR activation [37,38]. Dosage of EGFR components is known to maintain a biochemical balance, which dictates the final strength of the signaling pathway. Reducing Spitz levels in the background of Argos hypomorphic mutant reverted the rough eye phenotype to near-wildtype whereas in the background of Argos over-expression, Spitz dosage reduction enhanced the rough eye phenotype [18,25]. The relative strength of signaling decides the cellular choice, which in turn contributes towards proper pattern formation. But while relative expression dose of Spitz and Argos early in photoreceptor cell fate determination has been shown to be critical in patterning the *Drosophila* eye, the exact

identity of the cells expressing these genes or the degree of differential expression of these genes in the larval eye disc has been elusive.

Argos is both a negative regulator and a target of EGFR signaling. *argos* expression is a specific proxy for strong EGFR signaling [20,39]. To quantitatively understand the level of expression of diffusible EGFR components (Spitz and Argos), following their protein products can be problematic. Immunofluorescence staining or enhancer trap lines to track diffusible proteins is challenging as quantitative inference on the type of cells which have secreted them is not possible [39]. The dual phosphorylated ERK (dpERK) staining is classically used to read the strength of the EGFR pathway. Since most of the RTK pathways converge on the MAPK pathway, dpERK staining may not specifically and quantitatively report on EGFR strength. These problems can be circumvented by detecting endogenous mRNA *in situ*, which additionally could also indicate the identity of cells that express these genes. In recent years Single-molecule Fluorescence *In Situ* Hybridization (smFISH) has emerged as a powerful method to sensitively and quantitatively report the expression of more than one gene simultaneously in wholemount tissues, along with cell-to-cell variability [40–42]. We have previously shown that such methods can be used to detect even low levels of gene expression in the wholemount *Drosophila* tissues [40].

In this paper, using smFISH for EGFR pathway genes, we reveal an intriguing differential expression of *spitz* and *argos* in photoreceptor and non-photoreceptor cells of the larval eye disc. We show that relative expression levels of *spitz* and *argos* is important for the generation of proper ommatidial pattern rather than their absolute expression level. By systematically tuning the expression of EGFR pathway genes, we analyze the biochemical buffer range where the relative gene expression levels of the ligand *spitz* and the negative feedback regulator *argos* contribute towards pattern formation during the *Drosophila* eye development.

## Results

### EGFR signaling is directional during morphogenetic furrow progression

EGFR and the principal EGF, Spitz in *Drosophila* are expressed uniformly throughout multiple tissues during the development [43]. In the 3rd instar eye disc, Spitz is responsible for activation of EGFR signaling posterior to the morphogenetic furrow. Downstream targets of EGFR signaling like *argos* are expressed as a downstream response to PntP1 activation [44]. As Spitz and Argos are diffusible factors that can act in both a cell autonomous and non-autonomous manner and act in a 1:1 stoichiometry to modulate EGFR signaling [37], we reasoned that their relative expression may be key for tissue patterning. We investigated their transcription status quantitatively in cells behind the morphogenetic furrow in larval eye discs. In-situ hybridization (ISH) for *spitz* has been performed in eye discs at this stage [9], and patterns of dpERK signal also described by immunofluorescence [39]. Despite this, clear cell type-specific differences have not been highlighted most likely due to lower sensitivity of the methods used. We therefore performed smFISH (S1 Fig) to quantify *spitz* and *argos* mRNA numbers combined with immunofluorescence for Elav, a pan-neuronal marker [45] to distinguish photoreceptor (PR) cells from other non-photoreceptor (non-PR) and undifferentiated pools of cells. Elav staining revealed a rosette-like pattern of DAPI stained nuclei in PR cells that were used for RNA counts (S2A Fig). Since the nuclei are densely packed and individual cells are hard to segment in the eye imaginal disc, we quantified the mRNA expression on 3D stacks and normalized to a $1000\mu m^3$ tissue volume after delineating PR cells from non-PR cells. The membranes between PR and non-PR nuclei could be separated using Dlg, a membrane marker (S2B Fig). To our surprise we found that *spitz* is clearly expressed in higher levels in the photoreceptor cells (Elav-positive cells) compared to neighboring non-photoreceptor cells (Figs 1A, 1B, 1C and S1B). The

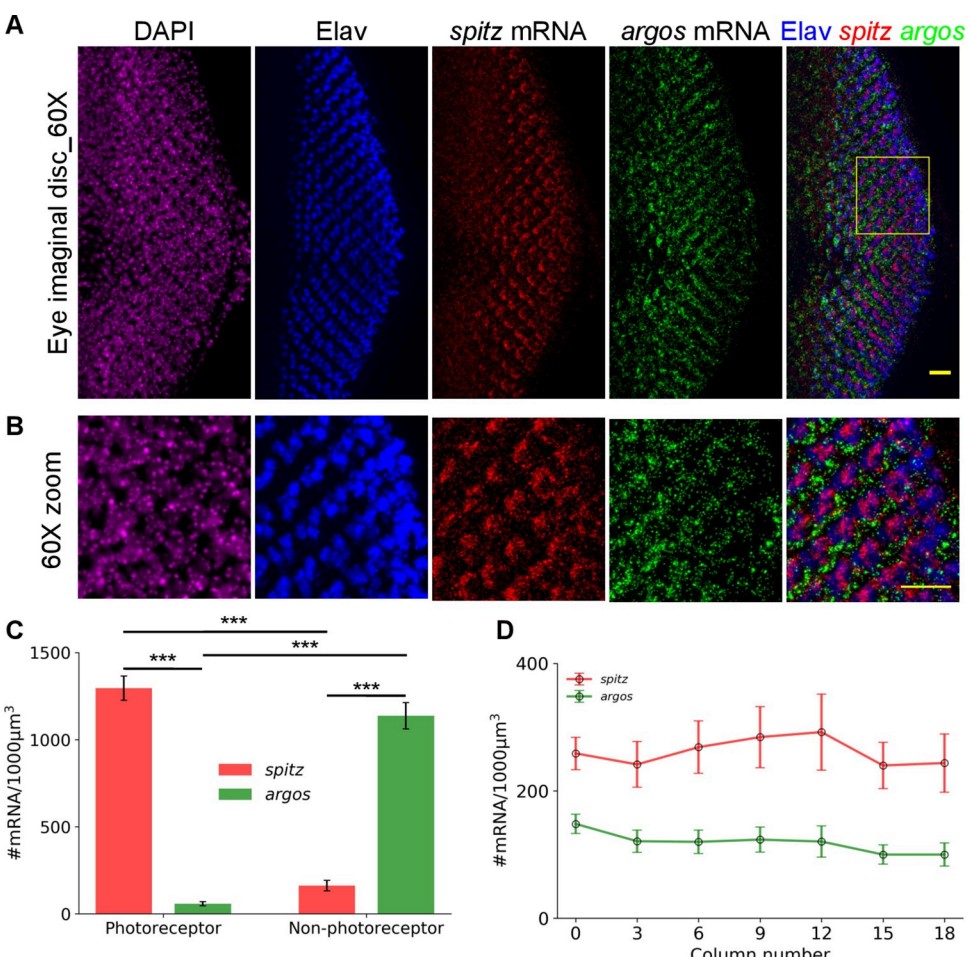

**Fig 1. EGFR signaling is directional in the eye imaginal discs.** (A) 60X images of 3rd instar eye imaginal discs stained with DAPI, Elav (Immunofluorescence; pan-neuronal marker), *spitz* mRNA and *argos* mRNA (single molecule RNA FISH). *spitz* mRNA is highly expressed only in the Elav-positive photoreceptor cells, whereas *argos* mRNA is highly expressed in the neighbouring non-neuronal cells. To clearly see the exclusive expression of *spitz* and *argos*, the image is zoomed into the yellow box and presented in (B). Scale bar is 10µm in (A) and 5µm in (B). These are z-projected images and hence individual transcripts get merged. A single z-slice is shown in S1B Fig. (C) The individual transcript molecules are counted in 3D using the StarSearch software (https://www.seas.upenn.edu/~rajlab/StarSearch/launch.html). The respective counts are represented per 1000µm³ of tissue volume as individual cells are hard to segment in this dense tissue. Photoreceptor (PR) and non-photoreceptor (non-PR) cells are identified by nuclei position corresponding to Elav as shown in S2A Fig. *spitz* expression is significantly higher in PR cells compared to non-PR cells whereas *argos* expression is significantly higher in non-PR cells compared to PR cells. (*** indicates p-values < 0.001 in a Student's t-test) (N = 9 tissues) (D) The mRNA counts are also represented in each photoreceptor column along a line from morphogenetic furrow (column 0) to the posterior end of the eye imaginal disc, as indicated in S2C Fig. (N = 8 tissues), irrespective of cell type (PR or non-PR)–overall mRNA counts are relatively constant along the line, i.e., with time after photoreceptor specification. Error bars in (C) and (D) are standard errors of mean. The images within the figure panels are created by the authors. Asterisks denoting significance of observed changes have been added to relevant graphs.

negative feedback regulator, *argos*, on the other hand, is expressed exclusively in the non-photo-receptor cells that have not yet made a cell fate choice. Because *argos* expression is direct target of EGFR signaling, *argos* expression in neighboring cells indicates an exclusively cell non-auton-omous effect of Spitz secreted by photoreceptor cells on their neighboring cells. The directional-ity in signaling can also be visualized by dpERK staining, a classical marker for high EGFR signaling, in the non-PR cells, and like *argos* mRNA higher dpERK levels are seen in non-PR

cells (S2C Fig). mRNA counts quantified along a line from the morphogenetic furrow to the posterior end of the eye imaginal disc do not show a large variation along the anterior to posterior axis (Figs 1D and S2D) indicating that the expression patterns are stable in time, because morphogenetic furrow progression is a proxy for time in development. We use eye discs from larvae of similar age (i.e. similar number of columns posterior to the furrow), but this indicates that otherwise too, the *spitz* and *argos* numbers are stable posterior to the furrow.

## Modulating absolute transcript numbers of *spitz* and *argos* may not translate to an effect on eye phenotype

Knocking down components of the DER pathway during development in the eye imaginal disc is known to affect the adult eye phenotype. Hypomorphic allele of *argos* and *argos* overexpression both give rise to rough eye phenotypes [18]. Halving Spitz in the background of hypomorphic Argos reverted the rough eye phenotype to normal whereas in the background of Argos overexpression, it further enhanced the rough phenotype [18]. We used the UAS-Gal4 system [46,47] to knockdown *argos* and *spitz*. We used GMR-Gal4 [48] to drive UAS-dsRNA in all cells posterior to the morphogenetic furrow and Elav-Gal4 [49] to drive UAS-dsRNA in photoreceptor (neuronal) cells respectively. smFISH showed the decrease in respective mRNA transcripts compared to wildtype CantonS eye discs. As expected, we observed decreased *argos* and *spitz* expression when GMR-Gal4 was used to drive *argos*-dsRNA and *spitz*-dsRNA respectively and the adult flies showed a fully penetrant rough eye phenotype (Fig 2A). To our surprise, while the Elav-Gal4 driver did show knockdown of the *spitz* expression when driving the *spitz*-dsRNA, there was absolutely no discernible defect in the phenotype of the adult eye (Fig 2A). The defect in the adult eye phenotype is not due to the GMR-Gal4 as the absolute *spitz* and *argos* transcript counts in the GMR/+ control (with and without heatshock at 29˚C) are similar to CantonS (S3A Fig), and the adult eyes show regularly patterned ommatidia (S3B Fig). *argos* and *spitz* mRNA was quantified in all the crosses along with CantonS and the numbers reflected the respective gene knockdowns (Fig 2B and 2C). The quantified gene expression could not explain the absence of effect on adult eye phenotype in the *spitz* knockdown driven by Elav-Gal4. As expected, eye discs did not show any change in transcript numbers when *argos* dsRNA was driven by Elav-Gal4, which is expressed exclusively in PR cells (S4A Fig), nor was there any defect in the adult eye (S4B Fig).

As relative dosage of EGFR pathway is known to be important for photoreceptor specification [18,25], we wondered if the ratio of expression of *spitz* to *argos* may be the critical determinant of the final eye phenotype instead of absolute levels of expression. This does assume that mRNA level expression translates directly to levels of final activated protein products, but given that Spitz and Argos act in a 1:1 stoichiometry, we thought this hypothesis is well worth testing. Thus, we analyzed the *spitz*-to-*argos* expression ratio through smFISH in the photoreceptor cells (Fig 2D) and eye fields with no distinction of PR and non-PR cells (Fig 2E). Remarkably, the *spitz*-to-*argos* ratio in the progeny of Elav-Gal4 with *spitz* dsRNA was similar to wildtype CantonS flies, while it was either significantly higher or lower in other genotypes (GMR-Gal4 driving *argos*-dsRNA and *spitz*-dsRNA). Thus, the gene expression ratio of the activating ligand, Spitz, and the negative feedback regulator, Argos, might dictate the final availability of free active ligand for pathway activation rather than their absolute expression levels. Cell-type specific transcript counting (PR cells) and full-field counting (no distinction of PR or non-PR cells) showed similar trends of *spitz*-to-*argos* ratios and we will describe full-field counts in all the further experiments.

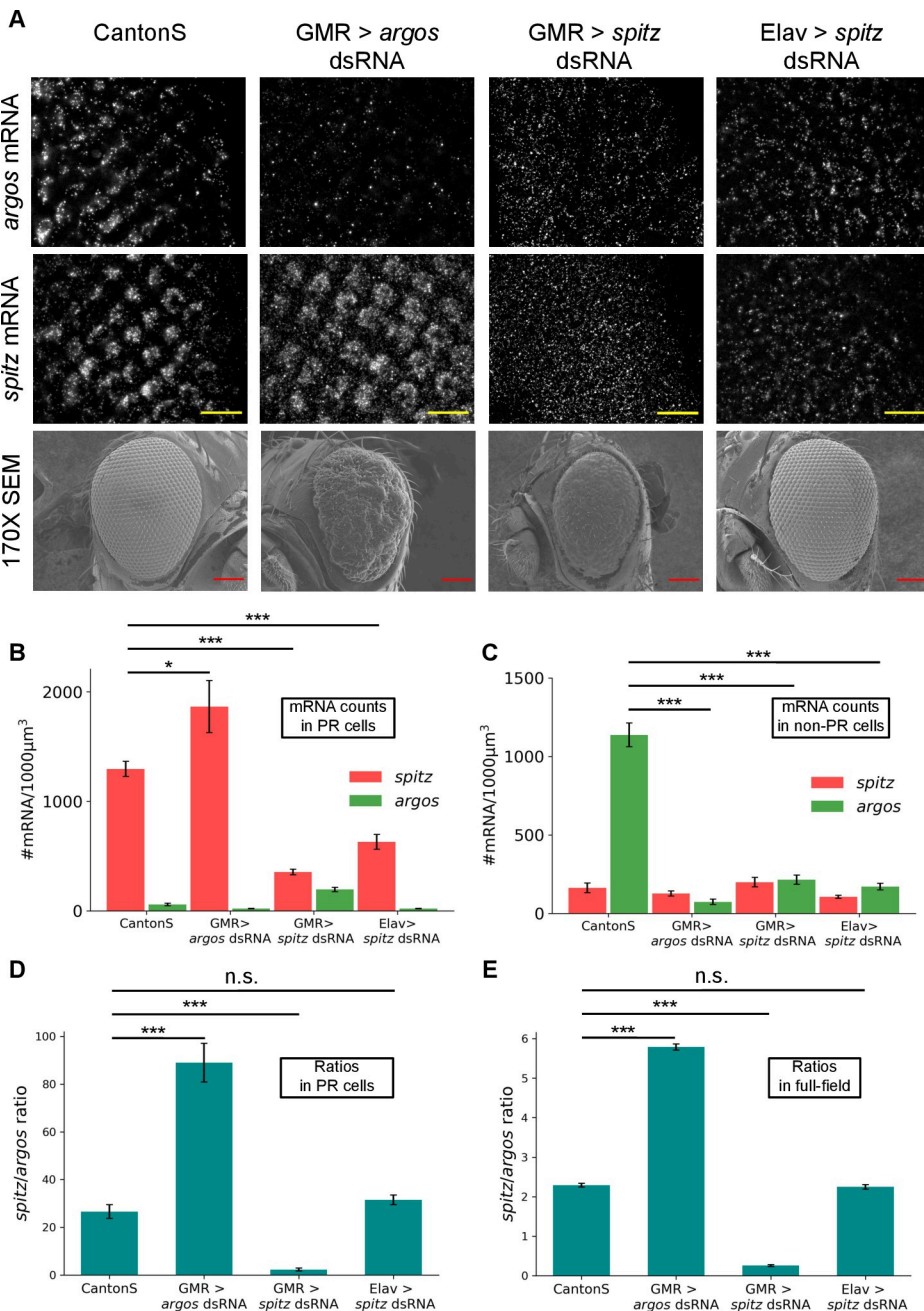

**Fig 2. *spitz*-to-*argos* ratio is important for proper ommatidial patterning.** (A) smFISH was performed in eye imaginal discs from represented crosses for *spitz* and *argos* mRNA in the two rows. In the third row, adult eye phenotypes are shown. The Elav > UAS *spitz* dsRNA progeny shows perfectly arranged ommatidia like the wildtype CantonS, while the other two crosses show rough eye phenotypes. Scale bar for widefield fluorescence images is 10µm and SEM images is 100µm. (B) and (C) mRNA numbers for *spitz* and *argos* were counted in photoreceptors and non-photoreceptors in respective crosses. *spitz* in PR cells and *argos* mRNA in non-PR cells show variation in number as expected from the respective crosses (*spitz* expression has p-value > 0.05 (*) in a Student's t-test when compared between CantonS and GMR-Gal4 driving *spitz* dsRNA. Other comparisons of *spitz* expression in PR cells and *argos* expression in non-PR cells are highly significant with p-values < 0.001 (***) in a Student's t-test). *spitz*-to-*argos* mRNA ratio in the photoreceptor cells (D) was calculated as dosage was known to be important for ommatidial pattern formation (N = 9 tissues in all the crosses). Most remarkably, knocking down *spitz* in PR cells with a Elav driver, knocks down *argos* numbers too, and overall the ratio is unchanged. When a GMR driver is used to knockdown either *argos* or *spitz* in the full field, the ratio is higher or lower than wildtype. *spitz*-to-*argos* ratio in the eye field irrespective of cell type (PR or non-PR) did not show any difference and is used henceforth in the paper (E). *spitz*-to-*argos* ratios

(D and E) in the GMR-Gal4 driving *argos* dsRNA and *spitz* dsRNA are significantly different (p-values < 0.001 (***) in a Student's t-test) when compared to wildtype CantonS while there was no significant difference in progeny of Elav-Gal4 driving *spitz* dsRNA (n.s. indicates p-values > 0.05 in a Student's t-test). Error bars in (B), (C), (D) and (E) are standard errors of mean. The images within the figure panels are created by the authors. Scale bars in Scanning Electron Microscopy images have been provided from knowledge of pixel size and converting appropriately. Asterisks denoting significance of observed changes have been added to relevant graphs.

## Buffered *spitz*-to-*argos* ratio is important for proper ommatidial patterning

To test the hypothesis that the ratio of expression of *spitz*-to-*argos* is critical for determining the final eye phenotype we attempted to use other cassettes to modulate their levels and also to tune their expression in a systematic manner. EGFR[CA] is a constitutive active form of EGFR, and hence the downstream targets will be expressed throughout the development. We used a GMR-Gal4 driver to express this constitutive EGFR. As GMR expresses in all the cells posterior to the morphogenetic furrow [31], the patterned expression of *argos* is lost (Fig 3A). *spitz*-to-*argos* expression ratio was quantified for the EGFR[CA] progeny, the balancer control (non-EGFR[CA]) and CantonS larvae. The balancer control had *spitz*-to-*argos* ratio levels similar to the CantonS around 2–2.2, whereas the EGFR[CA] expressing larvae had decreased ratio around 0.5 given high expression of *argos* and dramatically smaller and rough eye in adults (Fig 3B and 3C). To investigate if patterning defects in the adult eye vary continuously with the *spitz*-to-*argos* ratio or show up beyond a specific threshold indicative of developmental buffering, we tuned the expression of *spitz* and *argos*. We generated a GMR-Gal4 line with a temperature sensitive Gal80 (Gal80[ts]) [50,51] to tune the expression of *spitz* and *argos*. Gal80[ts] inhibits Gal4 activity at 18°C but turns inactive at higher temperatures allowing the control on UAS-Gal4 mediated gene expression. EGFR[CA] was crossed with Gal80[ts]; GMR-Gal4 and shifted to 29°C for different time points. The larvae were processed for smFISH and the absolute gene expression for *spitz* and *argos* (Fig 3D) and the *spitz*-to-*argos* ratios (Fig 3E) were quantified. The corresponding SEM images of the adult eyes were compared. The adult eye phenotype started to show patterning defects when the *spitz*-to-*argos* gene expression ratios neared 1 upon 40 minutes of heat shock at 29°C (Fig 3F). The intermediate *spitz*-to-*argos* expression ratio did not lead to roughening of the adult eye, indicating the phenotypic effects start to appear beyond a critical biological range.

## Threshold switch of *spitz*-to-*argos* ratio regulates phenotype

To recapitulate the buffer range with another gene, we crossed UAS-*argos* to the Gal80[ts]; GMR-Gal4 driver line to overexpress a*rgos*. Absolute gene expression (Fig 4A) and ratio (Fig 4B) were quantified. The corresponding SEM images (Fig 4C) were observed for patterning defects. The adult eyes showed patterning defects when the *spitz*-to-*argos* ratio neared 1. This confirmed that the adult eye phenotype is indeed contingent on the *spitz*-to-*argos* ratio. The phenotype did not show any defects in the adult eye even though the *spitz*-to-*argos* ratio dropped from 2.2 in the no-heat shock control till it reaches 1. We did not capture any defects in the ommatidial patterning in the flies with intermediate ratio. This suggests that patterning defects show up only when the *spitz*-to-*argos* ratio crosses a certain threshold and the intermediate ratios are buffered in the development. Having investigated the effects of decreasing the *spitz*-to-*argos* ratio, we wondered how far the buffering will persist when we increased it from wildtype levels. For this, we expressed an EGFR[DN] (dominant negative allele) with the Gal80[ts]; GMR-Gal4 driver, where ratios would be driven to values greater than wildtype. *spitz* and

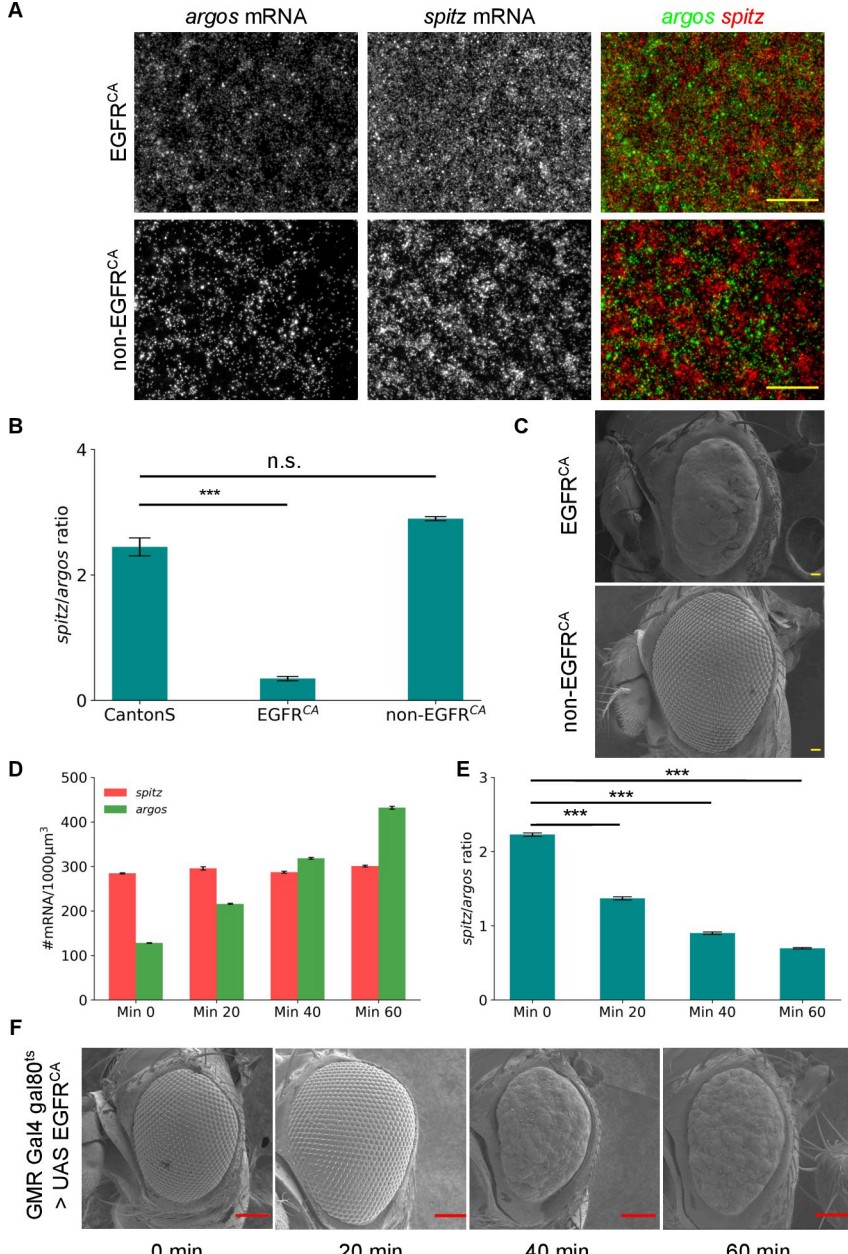

**Fig 3. Overexpression of EGFR completely disrupts ommatidial pattern from the adult eye.** (A) Eye imaginal discs of GMR > UAS EGFR$^{CA}$ progeny stained for *spitz* and *argos* mRNA. From the same cross, EGFR$^{CA}$ and non-EGFR$^{CA}$ larvae were distinguished using the balancer phenotype. Scale bar for widefield images is 10μm. Exclusive expression of *spitz* and *argos* is lost in EGFR$^{CA}$ which leads to change in the *spitz*-to-*argos* ratio represented in (B) [CantonS N = 9, EGFR$^{CA}$ N = 10, non-EGFR$^{CA}$ N = 9] (p-values from a Student's t-test; *** = < 0.001, n.s. = > 0.05). (C) 170X SEM images of GMR > UAS EGFR$^{CA}$ progeny. The ommatidial pattern is lost when EGFR is constitutively active and the size of the eye is also reduced. Heatshock was administered for 6 hrs to EGFR$^{CA}$ and non-EGFR$^{CA}$ larvae for maximal activation of the cassette. Scale bar is 10μm. (D) Absolute *spitz* and *argos* mRNA numbers from Gal80$^{ts}$; GMR-Gal4 > UAS EGFR$^{CA}$ progeny after different timepoints of heatshock at 29˚C. N = 8 for all time points. (E) *spitz*-to-*argos* ratios in eye discs corresponding to different time points of heat shock at 29˚C. Quantification of absolute mRNA numbers and ratios represented in all the above plots were calculated in the eye field irrespective of the cell type. Error bars in all the plots are standard errors of mean. The change in *spitz*-to-*argos* ratio with heatshock is significantly different irrespective of timepoints when compared with no-heatshock control (p-values < 0.001 (***) in a Student's t-test). (F) Adult eyes corresponding to different time points of heat shock at 29˚C. Scale bar is 100μm. No rough eye phenotype is seen for 20 min of heatshock, though a difference in the *spitz*-to-*argos* ratio is already seen at this point. Beyond this the eyes are fully rough, and the phenotype is fully penetrant. This is indicative of

developmental buffering. The images within the figure panels are created by the authors. Scale bars in Scanning Electron Microscopy images have been provided from knowledge of pixel size and converting appropriately. Asterisks denoting significance of observed changes have been added to relevant graphs.

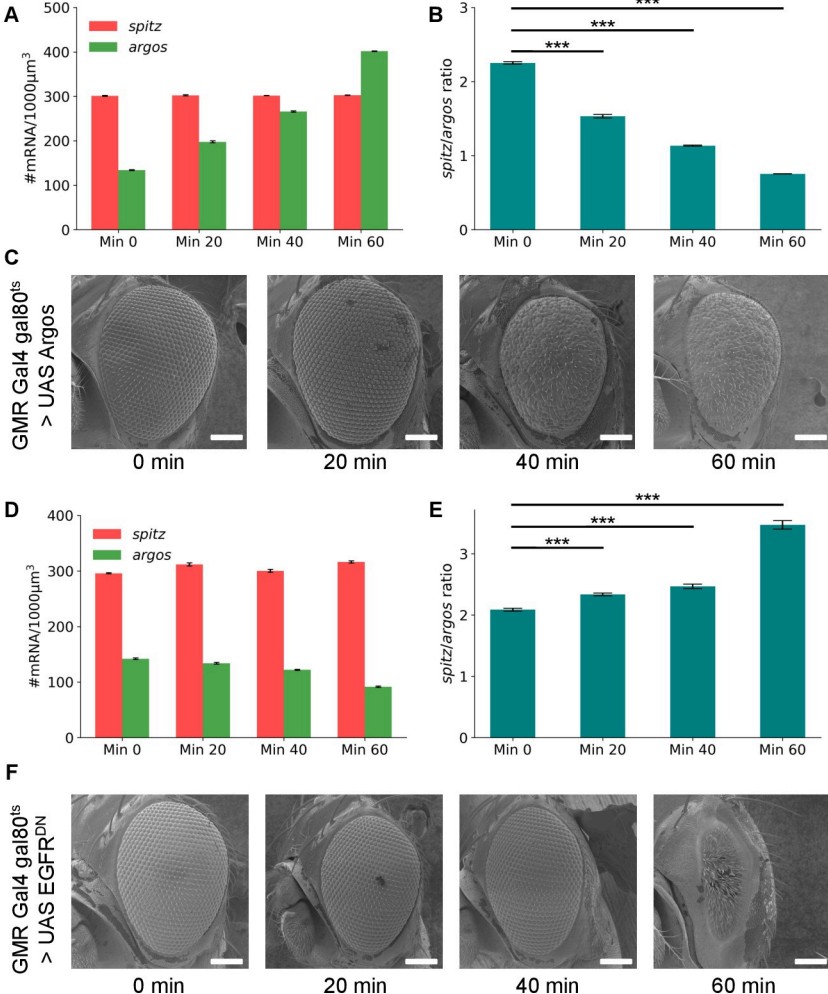

**Fig 4. Binary switch in *spitz*-to-*argos* threshold range for proper patterning of ommatidia.** (A) Absolute *spitz* and *argos* mRNA numbers from Gal80ts; GMR-Gal4 > UAS Argos progeny after different time points of heat shock at 29°C. (B) *spitz*-to-*argos* ratios from eye discs of Gal80ts; GMR-Gal4 > UAS Argos progeny after different time points of heat shock at 29°C. N = 9 tissues for all time points. The change in *spitz*-to-*argos* ratio with heatshock is significantly different irrespective of timepoints when compared with no-heatshock control (p-values < 0.001 (***) in a Student's t-test). (C) 170X SEM images of adult eyes of Gal80ts; GMR-Gal4 > UAS Argos progeny at different time points of heat shock at 29°C. (D) Absolute *spitz* and *argos* mRNA numbers from Gal80ts; GMR-Gal4 > UAS EGFRDN progeny after different time points of heat shock at 29°C. (E) *spitz*-to-*argos* ratios from eye discs of Gal80ts; GMR-Gal4 > UAS EGFRDN progeny after different time points of heat shock at 29°C. N = 8 tissues for all time points. Quantification of absolute mRNA numbers and ratios represented in all the above plots were calculated in the eye field irrespective of the cell type. Error bars for all the plots are standard errors of mean. The change in *spitz*-to-*argos* ratio with heatshock is significantly different irrespective of timepoints when compared with no-heatshock control (p-values < 0.001 (***) in a Student's t-test). (F) 170X SEM images of adult eyes of Gal80ts; GMR-Gal4> UAS EGFRDN progeny at different time points of heat shock at 29°C. Scale bar for all SEM images is 100μm. In both cases, the final adult phenotype is robust to certain variation in the ratio threshold, and breaks down beyond resulting in fully penetrant rough eye phenotypes. It should be noted that qualitatively the roughness phenotype looks different between UAS Argos and UAS EGFRDN. The images within the figure panels are created by the authors. Scale bars in Scanning Electron Microscopy images have been provided from knowledge of pixel size and converting appropriately. Asterisks denoting significance of observed changes have been added to relevant graphs.

*argos* mRNA transcripts were quantified (Fig 4D) along with their ratio (Fig 4E) at different time points of heat shock at 29˚C. The respective SEM images (Fig 4F) showed ommatidial defects when the ratio approached 3. The buffer range of *spitz*-to-*argos* for contributing towards proper ommatidial patterning might lie between 1–3 while the wildtype ratio lies between 2–2.2. While the term 'rough eye' can allude to a broad range of phenotypes, the patterning defects in the adult eye were quite different at both ends of the *spitz*-to-*argos* buffer range. Low ratios broadly showed irregularly spaced ommatidia with decreased number of bristle cells and high ratios showed bristle cells with complete loss of ommatidial clusters (Figs 3F, 4C and 4F). The differences could arise due to the effect of receptor signal strength on the photoreceptor cell fate specification or cellular division in the disc. We also used Elav-Gal4 to drive UAS Argos to ectopically overexpress *argos* in the photoreceptor cells (S5A Fig). The absolute levels of *argos* mRNA increases (S5B Fig) contributing to lower *spitz*-to-*argos* expression ratio (S5C Fig). However, the ratio remained around 1.4, due to increased *spitz* expression, and the lower expression ratio in Elav-Gal4 driving UAS-Argos in eye discs did not contribute to a rough eye phenotype in the adult (S5D Fig), reiterating the importance of buffered EGFR signaling in the developing eye.

## Different genetic perturbations affect proximate EGFR signaling outcomes differently

While EGFR signaling is important for photoreceptor specification in the larval stage, further EGFR-dependent identity determination also happens in the pupae; this finally results in a patterned eye phenotype in the adult, which is a distal outcome. To test the proximate signaling mechanisms that give rise to the final 'rough eye' phenotypes in the adults with different perturbations of EGFR signaling, we compared wildtype CantonS eye discs with discs from the progeny of Elav-Gal4 driving UAS-*spitz*-dsRNA and GMR-Gal4 driving UAS-*spitz*-dsRNA, UAS-EGFR[CA] and UAS-EGFR[DN] (Fig 5A). Yan, a negative transcription factor downstream to the EGFR pathway is known to be present in higher levels in the eye disc when EGFR is active [36]. High Yan in our experiments may be indicative of more sustained EGFR activity and hence was quantified. Yan intensity in discs of CantonS and progeny of Elav-Gal4 driving *spitz* dsRNA were similar compared to decreased intensity in discs from GMR-Gal4 driving *spitz* dsRNA and EGFR[DN]. Eye discs from GMR-Gal4 driving EGFR[CA] have significantly higher Yan compared to wildtype (Fig 5B). Previous studies have indicated a stochastic pulsatile behavior in EGFR signaling [52] and a higher Yan staining in a wider number of cells may indicate a longer or more frequent pulse duration for EGFR signaling, and clearly EGFR[CA] and EGFR[DN] affect this in opposite directions compared to wildtype. Further, we stained for neuronal markers as a proxy for proper differentiation of the photoreceptors. In wildtype eye discs, Elav is expressed in all PR cells and Prospero, under the control of EGFR, is expressed in R7 cell only [53,54]. This differentiation pattern is captured in CantonS and Elav-Gal4 driving *spitz* dsRNA discs. UAS-*spitz*-dsRNA driven by GMR-Gal4 showed a loss of neuronal cells, with very low signal from the Elav antibody. The residual Elav signal could be from the R8 founder photoreceptor neuron whose specification is not under the control of EGFR pathway. GMR-Gal4 driving EGFR[CA] shows more than one Prospero-positive cell per ommatidial cluster, whereas GMR-Gal4 driving EGFR[DN] shows complete loss of Prospero-positive R7 cells (Fig 5A). The number of Prospero-positive cells per ommatidial cluster in the above crosses is quantified (Fig 5C). We also stained the discs with EdU to capture the re-entry of G1 arrested cells in the morphogenetic furrow into the second mitotic wave. A band of S-phase cells just beyond the morphogenetic furrow is captured in CantonS discs along with discs from Elav-Gal4 driving *spitz* dsRNA. All crosses characterized by skewed ratios, showed complete loss of

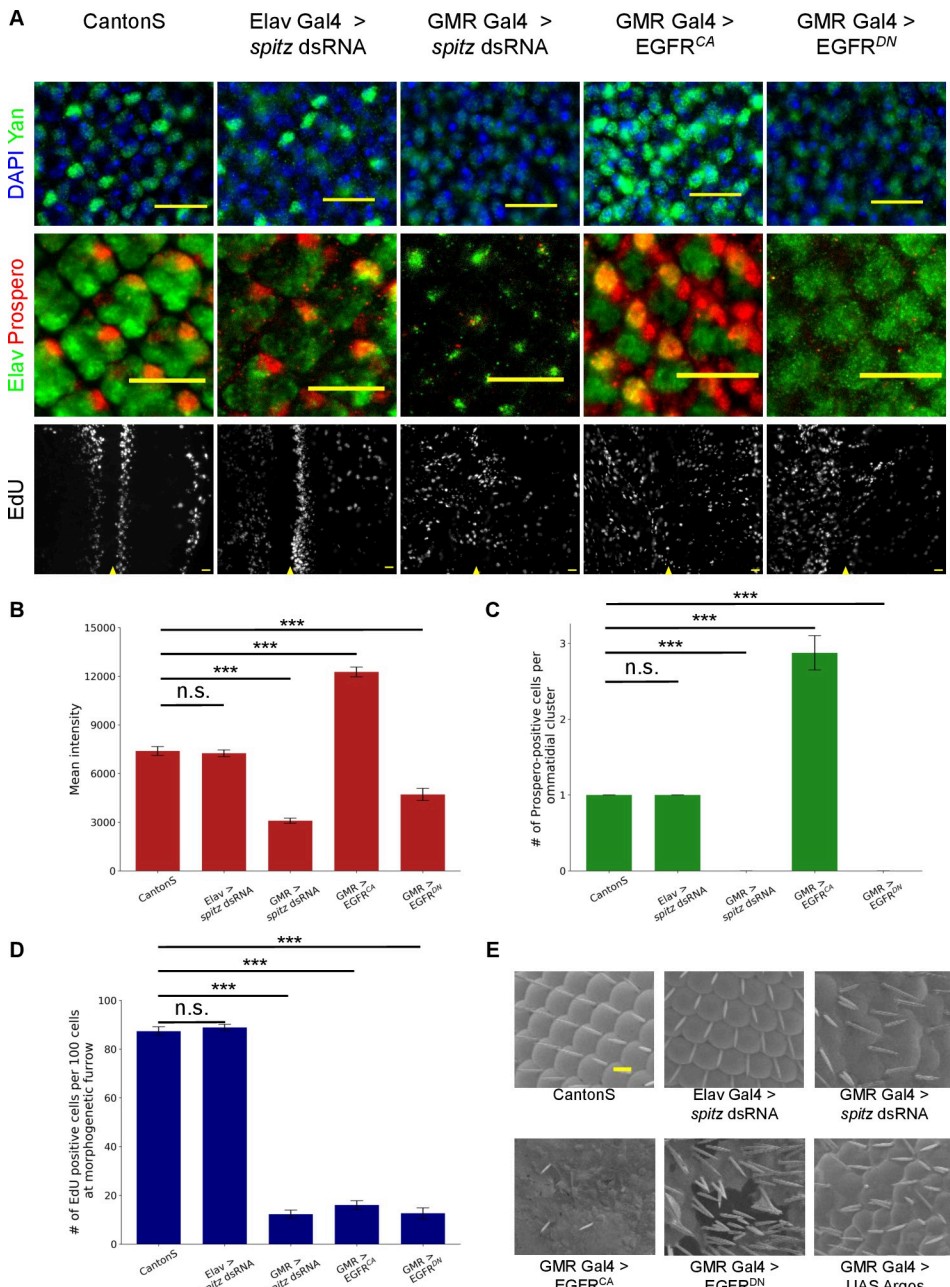

**Fig 5. Abnormal EGF receptor activity disrupts cell differentiation and cell division in the developing eye discs.**
Markers for proximate EGFR signaling (Yan), neuronal differentiation (Elav and Prospero) and cell division (EdU) are stained in eye discs from CantonS and discs with altered EGFR activity. (A) Yan intensity (first row) is similar in CantonS and Elav-Gal4 driving *spitz* dsRNA discs, whereas Yan intensity decreases when GMR-Gal4 drives *spitz* dsRNA or EGFR$^{DN}$ and increases with EGFR$^{CA}$. In CantonS eye discs, Elav stains all eight neuronal cells whereas Prospero stains only R7. The differentiation pattern is maintained in discs from Elav-Gal4 driving *spitz* dsRNA. Prospero-positive cells per cluster increased in discs when EGFR$^{CA}$ is driven by GMR-Gal4 whereas the signal was completely abrogated in discs when *spitz* dsRNA or EGFR$^{DN}$ was expressed. Eye discs were stained with EdU to visualize the band of S-phase cells posterior to the MF. Discs from CantonS and Elav-Gal4 driving *spitz* dsRNA showed the S-phase band which was absent when GMR-Gal4 was driving *spitz* dsRNA, EGFR$^{CA}$ and EGFR$^{DN}$. The intensity of Yan (B), number of Prospero-positive cells per ommatidial cluster (C) and number of EdU positive cells per 100 nuclei near morphogenetic furrow (D) are quantified. In all the quantifications, there is no significant difference between CantonS and progeny of Elav-Gal4 driving *spitz* dsRNA (p-values > 0.05 (n.s.) in a Student's t-test) and the difference with all crosses with GMR-Gal4 was significant (p-values < 0.001 (***) in a Student's t-test). Error bars in (B), (C) and (D) are standard errors of mean. (E) Zoomed-in SEM images showing ommatidial clusters and interommatidial

bristles cells in different crosses. Scale bar in all the images is 10μm. The images within the figure panels are created by the authors. Scale bars in Scanning Electron Microscopy images have been provided from knowledge of pixel size and converting appropriately. Asterisks denoting significance of observed changes have been added to relevant graphs.

S-phase band, implying abnormal EGF receptor activity completely disrupts the second mitotic wave. The number of EdU-positive cells per 100 nuclei after morphogenetic furrow were scored (Fig 5D). Together these markers for cell proliferation, differentiation and EGFR activity could explain the variety of the rough eye phenotype in the above crosses (Fig 5E). The adult eye of wildtype CantonS fly have perfectly patterned ommatidial clusters and bristles cells. Elav-Gal4 driving *spitz* dsRNA do not have any effect on the adult eye phenotype. Adult eye from GMR-Gal4 driving *spitz* dsRNA has reduced ommatidial clusters with randomly distributed bristle cells. GMR-Gal4 driving EGFR$^{CA}$ has no ommatidial clusters and drastically reduced number of bristle cells, whereas EGFR$^{DN}$ has smaller eyes with an increased number of bristle cells and complete loss of ommatidial clusters. GMR-Gal4 driving UAS-Argos have non-patterned ommatidial clusters and bristle cells. High EGFR activity contributes to higher photoreceptor cell specification with loss in the expansion of uncommitted cell pool resulting in loss of non-neuronal cell types in the ommatidia like the interommatidial bristle cells (eye phenotype of EGFR$^{CA}$ flies). On the other hand, downregulation of EGFR activity inhibited photoreceptor cell specification leading to predominant bristle differentiation (eye phenotype of EGFR$^{DN}$ flies). This also indicates that beyond the *spitz*-to-*argos* ratio, other signaling factors regulate the final eye phenotype. In fact, GMR-Gal4 driving *spitz* dsRNA and EGFR$^{DN}$ are more similar in terms of Yan intensity and number of prospero-positive cells, while they drive the *spitz*-to-*argos* ratio in different directions. And while GMR-Gal4 driving *spitz* dsRNA and EGFR$^{CA}$ drive the ratio in the same direction, at the molecular level how these affect EGFR signaling is different between these two manipulations. The final adult phenotype are also somewhat different between these two (Fig 5E), though they are more similar than the bristle-rich EGFR$^{DN}$ phenotype. The tight range for wildtype and balancer controls is perhaps indicative of the importance of relative Spitz and Argos doses, but the system is developmentally buffered to thresholds beyond this wildtype range. We never saw mixed phenotypes, and the flies always showed either largely proper ommatidial arrangements or fully penetrant rough eye phenotypes.

## Discussion

Developmental pathways have evolved mechanisms to monitor positional information in order to generate reproducible organismal patterns. These pathways are robust and insensitive to small changes in individual processes involved. Spatial differentiation, where a population of cells undergo deterministic molecular differentiation, brings about spatial patterns [55]. Redundancy of mechanism and negative feedback are two ways in which reliability in pattern formation is brought about [56,57]. Lateral inhibition by diffusible molecules is another mechanism that can be used to generate patterns [58]. For systems which do not depend on developmental history, environmental makeup determines their molecular differentiation contributing towards generating a pattern [59].

In this paper, we propose a mechanism where relative expression levels of principal EGFR ligand, Spitz and negative feedback regulator Argos determines the extent of EGFR activation which is crucial for the periodic ommatidial pattern. Our data suggests that it is not the absolute gene expression but the balance between gene networks on the whole which may contribute towards pattern formation. GMR-Gal4 is expressed in all cells posterior to the morphogenetic furrow [48] and any expression cassette under the UAS is expressed strongly.

Whereas, Elav-Gal4 is expressed only in neuronal cells and the expression is strongest towards the posterior end of the eye disc [49]. While Elav-Gal4 expression occurs in differentiated neurons starting with R8 in the eye disc (for which EGFR signaling is not needed), but beyond R8 specification, Spitz and Argos levels are important for subsequent PR cell differentiation and also in the pupal stages [60]. Although we observe an equally drastic reduction in *argos* expression when UAS-*spitz* dsRNA driven by GMR-Gal4 and Elav-Gal4 (Fig 2C), the eye discs show reduced dpERK staining in both cases (S6A Fig). The reduction was lower when Elav-Gal4 driver was used, corresponding to the absence of phenotype in the adult eye (S6B Fig). The eye discs expressing EGFR[CA] construct under GMR-Gal4 Gal80[ts] with *spitz*-to-*argos* ratio near 1, showed a discontinuous S-phase band after the morphogenetic furrow indicating a lower population of cells entering the second mitotic wave (Fig 5). Fewer cells entering the second mitotic wave leaves the tissue field with fewer uncommitted cells to make cell fate decisions. This can affect pattern formation to a great extent. This could also explain fewer number of bristle cells in the rough adult eyes (Fig 3) as bristles cell fate is assigned from cells arising from the second mitotic wave [61]. It should also be noted that the rough eye phenotype for EGFR[DN] is rather different from EGFR[CA] and shows a profusion of bristles (compare 60 min time-points of Figs 4F and 3F). This mechanism of relative expression determining phenotype supports older work on the importance of *spitz*-to-*argos* dose as a critical determinant of eye patterning [18,25]. Indeed mRNA levels may not be always predictive of protein levels, and both translational regulation and post-translational modifications may well affect biological function. However, under conditions of stress [62] or where specific transcriptional programmes bring about developmental outcomes, transcript levels may be thought to be well-correlated to protein levels. In addition, experiments using smFISH lets us clearly identify the cells that are expressing specific genes, where the diffusible protein end-products may not provide as conclusive answers. We did attempt performing antibody staining for Spitz and Argos but such relative measures cannot be used to comment on expression stoichiometry (S7 Fig). Our study shows a clear differential expression of *spitz* and *argos* mRNA in the early eye field contributing to photoreceptor fate determination and also addresses the sensitivity of the system to the heterogeneity in the expression levels of gene networks and makes developmental programs robust. It has to be noted of course that signaling via EGFR is not the only pathway determining the ommatidial pattern in the eye. For example, Notch is known to play an important role in the initiation of neural development and also in ommatidial rotation [63,64]. Buffered regulation of genes in different developmental pathways that crosstalk can decrease sensitivity to variations in a gene network and can help explain other reproducible and stereotypical patterns generated throughout the development.

## Materials and methods

### Drosophila stocks and crosses

All fly strains were grown on standard cornmeal agar at 25˚C. CantonS line was used as the wildtype fly. The cassettes used for modulating the EGFR pathway were UAS-*argos* dsRNA [65], UAS-*spitz* dsRNA [66], UAS-EGFR[CA] (FBst0305944), UAS-Argos (FBst0005363) and UAS-EGFR[DN] (FBst0005364). Tissue specific GMR (FBst0300831) and Elav (FBst0008765) drivers were used to express the above constructs. Heatshock to activate GMR-Gal4 and Elav-Gal4 was given to late 2[nd] instar larvae overnight in 29˚C incubator. Gal80[ts]/FM7 (FBst0007016) was used to generate Gal80[ts]; GMR-Gal4 with a temperature sensitive Gal80. The crosses with Gal80[ts] line were set at 18˚C and shifted to new vials every alternate day. Towards the end of the 2[nd] instar, the larvae were shifted to 29˚C for particular time points and immediately shifted back to 18˚C and continued to grow. Larvae were collected for

dissection around mid-to-end 3rd instar and processed for smFISH. Adult progeny from these crosses were allowed to emerge at 18˚C to assess the effect on ommatidial patterning through SEM.

## Tissue preparation and smFISH

3rd instar larvae were collected in nuclease-free 1X PBS (Ambion, AM9624) and washed once with the same. The larvae were flipped and transferred to a microcentrifuge tube containing 4% Paraformaldehyde (PFA, Sigma, P6148) in 1X PBS for 25 minutes at room temperature. The fixative was aspirated and washed twice with 1X PBS. The tissue was permeabilized using 0.3% Triton-X 100 (Sigma, T8787) in 1X PBS for 45 minutes at room temperature. The permeabilizing agent was aspirated and the tissues were washed twice with 1X PBS. The tissues were kept in 70% ethanol at 4˚C overnight. The tissues were washed twice with a wash buffer (20% Formamide (Ambion, 9342) and 2X SSC (Ambion, AM9763) in nuclease-free water) for 30 minutes each at 37˚C. The wash buffer was aspirated and the tissues were incubated with a hybridization mix overnight at 37˚C. The probe sequences for *spitz* and *argos* is given in the previous paper [40]. The mix was removed and tissues were washed twice with wash buffer at 37˚C. DAPI (Invitrogen, D1306; 2µg/ml) in wash buffer was added to the tissues and incubated for 30 minutes at 37˚C. The tissues were then washed with 2X SSC twice for 5 minutes. The eye imaginal discs were dissected in 2X SSC and mounted on a clean slide with a drop of Vectashield (Vector Labs). A cartoon depicting the brief protocol and singly labeled 20-mer oligonucleotide probes binding to mRNA target is shown in S1A Fig. StarSearch software (https://www.seas.upenn.edu/~rajlab/StarSearch/launch.html) was used to count transcripts in 3D as done before [40]. The absolute counts and ratios were calculated in regions of interest on the eye field and then averaged.

## Antibody staining

The tissues were fixed and permeabilized as described earlier. The tissues were then washed twice with 1X PBS. Blocking solution (5% BSA in 1X PBS) was added to the tissues and incubated for 1 h at room temperature on a rotating mixer. Primary antibody (1:1000 Elav [Rat-Elav-7E8A10 anti-elav was deposited to the DSHB by Rubin, G.M.] [35], 1:500 Yan [anti-Yan 8B12H9 was deposited to the DSHB by Rubin, G.M.] [36], 1:500 Argos [Anti-Argos 85/2/16 was deposited to the DSHB by Freeman, M.], 1:500 Spitz [anti-Spitz was deposited to the DSHB by Shilo, B.-Z.], 1:1000 Prospero [Prospero (MR1A) was deposited to the DSHB by Doe, C.Q.] [67], 1:1000 dpERK [CST, 9101]) diluted in the blocking solution (5% BSA [Sigma, A2153] in 1X PBS) was added after 1hour and incubated overnight at 4˚C. 1:1000 secondary antibody (Goat anti-Rat 488 [Invitrogen, A11006], Goat anti-Rat 647 [Invitrogen, A21247], Goat anti-Mouse 594 [Invitrogen, A11032], Goat anti-Mouse 488 [Invitrogen, A11029], Goat anti-Rabbit 594 [Invitrogen, A11037]) diluted in blocking solution was added to the tissues and incubated for 3 hours at room temperature. Tissues were then washed twice with 1X PBS for 15 minutes each on a rotating mixer. DAPI in 1X PBS was added and incubated for 30 minutes. The tissues were finally washed twice with 1X PBS for 5 minutes each.

For simultaneous smFISH-IF, the primary antibody is added into the hybridization mix and incubated overnight at 37˚C. The secondary antibody was diluted to 1:1000 in nuclease-free blocking solution and incubated with tissues for 3 hours at room temperature on a rotating mixer.

## Image acquisition

The tissues were imaged on an Olympus BX63 upright widefield fluorescence microscope with a Retiga 6000 (Qimaging) CCD monochrome camera. The slides were kept at 4˚C for 1 hour at least before imaging. The images were acquired using a 60X, 1.42 N.A. oil immersion objective with a z-step size of 0.3μm. Narrow band-pass filters (ChromaTechnology—49309 ET-Orange#2 FISH for Quasar 570 labelled probes; 49310 ET- Red#2 FISH for CAL fluor 610 labelled probes) were used to spectrally separate single transcripts imaged in two colors.

## Analysis of mRNA counts in the 3rd instar eye disc

The smFISH protocol works perfectly in the wholemount tissues (~40μm). 30–40 z-slices corresponding to the photoreceptor focal planes (based on nuclear arrangement of DAPI-stained PR cells) are separated from the raw image stack and used for representation and analysis unless otherwise mentioned. For representation, the stacks were z-projected, merged wherever necessary. For mRNA counts, StarSearch software was used as mentioned earlier. The ROIs were drawn free-hand, area roughly in the order of 30000–35000 pixels. The counts are analyzed in 3D stack and converted to volume as described in our previous paper [40]. For counts along the A-P axis of the tissue, ROIs were drawn from the morphogenetic furrow to the posterior end. Equal number of columns were analyzed and represented in the all the analysis and henceforth narrow-window staging of larvae was not necessary. All numerical data underlying the graphs are provided in the S1 Data.xlsx file. GraphPad (https://www.graphpad.com/quickcalcs/ttest1/) was used to calculate p-values in a Student's t-test for statistical significance.

## EdU labeling using Click-chemistry

3rd instar larvae were flipped and incubated in Schneider's insect medium [Sigma, S0146] containing 10μM 5-Ethynyl-2'-deoxyuridine (5-EdU) [Jena Bioscience, CLK-N001] for 30 minutes. The larvae were then fixed and permeabilized as described earlier. Larvae were then incubated with 250μl click reaction cocktail (29μl 100mM ammonium guanidine, 10μl 100mM copper sulfate, 10μl 50mM THPTA [Sigma, 762342], 25μl freshly made 1M sodium ascorbate and 1μl 6.2mM Azide-fluor 488 [Sigma, 760765] or Cy5-Azide [Sigma, 777323] in 175μl HEPES buffer) for 30 minutes. The larvae were then processed for immunofluorescence or directly counter-stained with DAPI. Eye imaginal discs were then dissected and mounted in Vectashield.

## SEM imaging of adult eyes

Adult fly heads were dissected in 1X PBS. PBS was aspirated and 4%PFA was added for 25 minutes at room temperature. Fly heads were then washed twice with 1X PBS for 5 minutes each. Conducting carbon adhesive tape was spread on the SEM specimen stub. The fly heads were aligned on the carbon tape and were air-dried. The images were captured using a lower electron detector in a JEOL JSM7200F microscope at 170X magnification.

## Supporting information

**S1 Fig.** (A) Cartoon showing single molecule RNA FISH protocol. Tissues are fixed and permeabilized and hybridized overnight with complementary probes. Multiple 20 nt long oligos each carrying a 3' fluorophore is used to decorate the mRNA of interest following previous protocols. Singly-labeled 20-mer probes bind along the mRNA length and appear as diffraction-limited spots when imaged. Scale bar is 5μm. (B) Zoomed images of a single z-slice showing single transcripts of *spitz* mRNA and *argos* mRNA. Scale bar is 5μm. The images within

the figure panels are created by the authors.
(TIFF)

**S2 Fig.** (A) mRNA was counted specifically in the photoreceptor using the rosette like arrangement of PR nuclei. The marked yellow region corresponds to one photoreceptor cluster marked in DAPI channel which also stains positive for a pan-neuronal marker, Elav. (B) CantonS 3rd instar larval eye disc stained with Dlg and Elav antibody. The nuclei around the clusters of Elav-positive neuronal photoreceptor (PR) cells, are the non-photoreceptor (non-PR) cells. The non-PR cells are also separated by Dlg which stains the membrane. (C) The Elav-positive cells and cells stained with dp-ERK are exclusive, again saying that EGFR signaling is activated in the neighbouring cells to the ligand source of PR cells. Scale bar is 10μm in (A) and 5μm in (B and C). (D) Cartoon representing the line along which *spitz* and *argos* mRNA counts were analysed (A: anterior end; MF: morphogenetic furrow; P: posterior end). The images within the figure panels are created by the authors.
(TIFF)

**S3 Fig.** (A) Absolute *spitz* and *argos* mRNA numbers are plotted for CantonS and GMR-Gal4 x CantonS flies with and without heatshock at 29˚C. The absolute mRNA numbers do not show significant difference (p-values > 0.05 in a Student's t-test). (N = 8 tissues for all genotypes) Error bars are standard errors of mean. (B) 170X SEM images of the respected genotypes all show perfectly patterned ommatidia. Scale bar is 100μm. The images within the figure panels are created by the authors. Scale bars in Scanning Electron Microscopy images have been provided from knowledge of pixel size and converting appropriately.
(TIFF)

**S4 Fig.** (A) Absolute mRNA counts of *spitz* and *argos* in photoreceptors and non-photoreceptors are plotted for CantonS and Elav-Gal4 driving *argos* dsRNA. There is no significant difference in absolute count of *spitz* mRNA in photoreceptors and *argos* mRNA in non-photoreceptors (p-values > 0.05 in a Student's t-test) (N = 9 tissues) Error bars are standard errors of mean. (B) The adult eye of flies with Elav-Gal4 driving *argos* dsRNA did not show any defects when compared to the wildtype adult eyes. Scale bar is 100μm. The images within the figure panels are created by the authors. Scale bars in Scanning Electron Microscopy images have been provided from knowledge of pixel size and converting appropriately.
(TIFF)

**S5 Fig.** (A) *spitz* and *argos* expression pattern in the eye imaginal disc is disrupted by overexpression of UAS Argos by Elav-Gal4 driver in the photoreceptor cells. (B) Absolute *spitz* and *argos* mRNA numbers from Elav-Gal4 driving UAS Argos are plotted along with wildtype CantonS. (C) *spitz*-to-*argos* ratios from the same eye discs are quantified. N = 8 tissues for all genotypes. Quantification of absolute mRNA numbers and ratios represented in all the above plots were calculated in the eye field irrespective of the cell type (*** indicates p-values < 0.001 in a Student's t-test). Error bars for all the plots are standard errors of the mean. The change in *spitz*-to-*argos* ratio is significantly higher in the Elav-Gal4 driving UAS Argos when compared to wildtype CantonS eye discs (p-values < 0.0001 in a Student's t-test). (D) 170X SEM images of adult eyes from the wildtype and Elav-Gal4 driving UAS Argos. Scale bar is 10μm in (A) and 100μm in (D). The images within the figure panels are created by the authors. Scale bars in Scanning Electron Microscopy images have been provided from knowledge of pixel size and converting appropriately. Asterisks denoting significance of observed changes have been added to relevant graphs.
(TIFF)

**S6 Fig.** (A) dpERK along with Elav is stained in eye discs from CantonS, Elav-Gal4 driving *spitz* dsRNA and GMR-Gal4 driving *spitz* dsRNA. Unlike smFISH which affords absolute mRNA counts, immunofluorescence is a relative measure and it is more difficult to compare across experiments in different strains. But under identical staining and imaging conditions, both the crosses seemed to show significantly reduced dpERK staining compared to CantonS. The reduction was lower for the Elav driver, corresponding to the absence of a phenotype in the adult eye. Residual dpERK signal is observed around the Elav-positive neuronal cells with Elav-Gal4 driving *spitz* dsRNA. In GMR-Gal4 driving *spitz* dsRNA, patterned Elav and dpERK staining is lost. (B) Mean Intensities of dpERK in a fixed ROI is plotted here clearly showing the decreased dpERK staining when Elav or GMR-Gal4 drives *spitz* dsRNA. (*** indicates p-values $< 0.001$ in a Student's t-test). N = 6 in all genotypes. Error bars are standard errors of mean. Scale bar is 10µm. The images within the figure panels are created by the authors. Asterisks denoting significance of observed changes have been added to relevant graphs.
(TIFF)

**S7 Fig. CantonS 3$^{rd}$ instar eye discs are stained with Spitz (1:500 dilution Rat anti-Spitz, DSHB) and Argos (1:500 dilution Mouse anti-Argos 85/2/16, DSHB) antibodies.** Scale bar is 10µm. Since cleaved Spitz and Argos are diffusible molecules, the staining does not recapitulate the *spitz* and *argos* mRNA expression patterns exactly. Moreover, unlike single molecule FISH (smFISH) for RNA that yields absolute mRNA counts, immunofluorescence experiments show only relative changes. The signal intensities captured are dependent on the affinity of the antibodies, quantum yield of the fluorophores used in secondary detection, image acquisition parameters etc. and are different for the two antibodies used. Thus while for a given antibody relative changes can be followed, there is no easy way of comparing staining intensities for antibodies directed against different antigens. Therefore such staining cannot be used for determining expression ratios of Spitz and Argos, unlike with smFISH, which both clearly marks the source cells and also yields absolute transcript counts. To our mind this provides two clear examples of the power of smFISH. The images within the figure panels are created by the authors.
(TIFF)

**S1 Data. All numerical data underlying the graphs are provided in the S1 Data.xlsx file** (XLSX)

## Acknowledgments

We thank Dr. Rohit Joshi and Dr. Rakesh Mishra for the gift of fly lines. We also thank Shravani Anagandula for help with SEM imaging. We also acknowledge FlyBase for the informational resources [68].

## Author Contributions

**Conceptualization:** Nikhita Pasnuri, Krishanu Ray, Aprotim Mazumder.

**Data curation:** Nikhita Pasnuri.

**Formal analysis:** Nikhita Pasnuri.

**Funding acquisition:** Manish Jaiswal, Aprotim Mazumder.

**Investigation:** Nikhita Pasnuri, Aprotim Mazumder.

**Methodology:** Nikhita Pasnuri, Manish Jaiswal, Krishanu Ray, Aprotim Mazumder.

**Project administration:** Aprotim Mazumder.

**Resources:** Nikhita Pasnuri, Manish Jaiswal, Krishanu Ray, Aprotim Mazumder.

**Supervision:** Manish Jaiswal, Krishanu Ray, Aprotim Mazumder.

**Validation:** Nikhita Pasnuri.

**Visualization:** Nikhita Pasnuri.

**Writing – original draft:** Nikhita Pasnuri.

**Writing – review & editing:** Manish Jaiswal, Krishanu Ray, Aprotim Mazumder.

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
