## [Decision Letter · Decision Letter 0]

26 Oct 2022

Dear Dr Mazumder,

Thank you very much for submitting your Research Article entitled 'Buffered EGFR signaling regulated by spitz-to-argos expression ratio is a critical factor for patterning the Drosophila eye' to PLOS Genetics.

The manuscript was fully evaluated at the editorial level and by independent peer reviewers. The reviewers appreciated the attention to an important problem, but raised some substantial concerns about the current manuscript. Based on the reviews, we will not be able to accept this version of the manuscript, but we would be willing to review a much-revised version. We cannot, of course, promise publication at that time.

If you decide to revise the manuscript for further consideration at PLOS Genetics, please aim to resubmit within the next 60 days, unless it will take extra time to address the concerns of the reviewers, in which case we would appreciate an expected resubmission date by email to plosgenetics@plos.org.

We are sorry that we cannot be more positive about your manuscript at this stage. Please do not hesitate to contact us if you have any concerns or questions.

Yours sincerely,

Justin Kumar, PhD

Guest Editor

PLOS Genetics

Gregory P. Copenhaver

Editor-in-Chief

PLOS Genetics

Reviewer's Responses to Questions

**Comments to the Authors:**

Reviewer #1: This manuscript by Nikhita et al reports how EGFR activation is tuned by the relative expression levels of a activation ligand Spitz and a inhibitory regulator Argos to regulate the patterning of the Drosophila eye. By using smFISH to identify the cells that produce the ligands and by a number of genetic means to perturb EGFR signaling, the authors show that the ratio of spitz-to-argos expression is a critical determinant of the final adult eye phenotype, rather than absolute levels of ligands expression. They propose a mechanism in which the relative expression levels of Spitz and Argos determine the extent of EGFR activation, which is critical for periodic ommatidium patterns. The study is potentially interesting, but additional experiments and clarifications are needed to strength the conclusions.

Major points:

One key observation from the authors is that knocking down spitz in photoreceptor cells by elav-GAL4 does not cause any patterning phenotype, based on which the authors propose that the absolute level of spitz is not important. However, there could be additional EGFR ligands involved in the process, such as keren, which may have redundant functions with spitz. Therefore, the authors appear to make the conclusion based on the assumption that spitz is the only activation ligand involved in the process, which is probably not true.

Should the EGFR signaling activity be affected by either altering spitz or arogos? If spitz is the only ligand, depleting of spitz would cause the loss EGFR signaling.

From the SmFISH results, the authors found that spitz is mainly expressed in PR cells, and argos is exclusively expressed in non-PR cells. However, as the PR cells and non-PR cells are tightly associated with each other, and the cell boundaries are not labeled in the images, it remains possible that spitz and argos are expressed in the same types of cells but with distinct subcellular distributions. Co-staining with a cell membrane marker should help to clarify this issue.

Unlike Elav-GAL4, which drives gene expression in PR cells, the authors show that knock-down of spitz by GMR-GAL4, which drives gene expression in both PR and non-PR cells, do cause patterning defects. Should it be indicative that spitz is expressed in non-PR cells and is important for eye patterning?

The authors examined the ratio of spitz-to-argos in settings when EGFR signaling is perturbed by expressing EGFR-act or EGFR-DN. But since the effects of these genetic perturbations can no longer be affected by the signaling ligands, the examination of the ligand levels does not seem to be meaningful and informative to support the “buffered signaling” hypothesis.

Minor points:

Is the mRNA count in eye imaginal discs equal to that of PR cells plus non-PR cells? Fig. 2B and C show that spitz/argos is about 1, but this is not consistent with the results shown in Fig. 2D and E.

Fig. 5: Yan appears to be expressed in Elav+ cells. Does it indicate that EGFR signaling is active in PR cells? But as shown in Fig. 1 and Supplementary Fig. 2, dpERK is restricted to non-PR cells.

Reviewer #2: The manuscript presents data to further unravel the role of the Epidermal Growth Factor Receptor, EGFR, for pattern formation in the developing eye of Drosophila. In the eye, the two ligands Spitz and Argos act antagonistically to activate or repress EGFR, respectively. Therefore, expression of the two ligands has to be tightly regulated to ensure correct receptor activation and ultimately correct eye patterning.

Here, the authors used single molecule RNA Fluorescent in situ Hybridisation (smFISH) to quantify spitz and argos transcripts at cellular resolution in the eye discs of third instar larvae of various genetic backgrounds (wildtype, expression of dsRNA and overexpression). Thereby they demonstrate that spitz RNA is restricted to developing photoreceptor cells, while argos RNA is detected in adjacent, non-photoreceptor cells. They further demonstrate that it is not the absolute amount of these RNAs that determines the activation/inhibition of the EGFR pathway, but rather the ratio of the spitz/argos mRNA. Interestingly, the tissue can tolerate a rather wide range of this ratio without showing any defects. This indicates that the system is well buffered to allow proper development even upon small perturbations.

In the second part the authors analyze what they call “the proximal EGFR signalling outcome” to explain the variety of pattern defects achieved upon the different genetic perturbations. This part stays somewhat vague and less clear.

Overall, this manuscript provides novel and interesting results by showing that the ratio of the two ligands Spitz and Argos are important for appropriate EGFR signalling. These data are interesting for scientists interested in EGFR signalling in general. Results presented in the second part are less clear and certainly more interesting for scientists working on pattern formation in the Drosophila eye.

Specific comments:

1. Quantification of RNAs: According to Suppl. Fig. 2, they measured the RNA along a line from the morphogenetic furrow to the posterior part of the disc. Along this line, cells/ommatidia are in different stages of development, but they did not take this into account and write (line 142) that there is no “large variation along the anterior to posterior axis”. However, I have the impression that in Canton S there is a gradient of argos expression, being higher closer to the morphogenetic furrow.

The difference in developmental stage could be of importance when they overexpress UAS constructs using GMR-Gal4, which then affects cells in different developmental stages. This may also contribute to the variety of eye defects described in the second part of their manuscript.

2. Line 24 (and other places, e.g. line 87): what exactly do they mean with “early development”?

3. Materials and Methods, line 331-333: How did they stage 3rd instar larvae raised at 18°C? In addition, to inactivate Gal80, they shifted the larvae to 29°C for various times. I guess they prepared the eye discs immediately after the exposure to the higher temperature?

4. Line 271: here and in other places the authors use the word “PR cell differentiation”. I guess what they mean here (and in other places as well) is PR cell fate specification, since there are more PR cells and less uncommitted cells. I suggest to be more precise in these cases. (In fact they write: “… contributes to higher PR cell differentiation”, I guess they mean a higher number of PR cells … . ).

5. Line 178: They write that the “…. spitz to argos ratio ….. was either significantly higher or lower in other genotypes…”. How did they determine the significance?

Reviewer #3: This study focuses on the regulation of EGFR signalling pathway activation in the developing eye imaginal disc. The authors employed a smFISH technique to analyze and compare the transcript expression of, other way difficult to measure, the diffusible ligands of EGFR, spitz and argos. Their results reveal that whereas spitz is expressed specifically in photoreceptors cells, argos expression is detected in the surrounding non differentiated cells. This interesting observation suggests that EGFR signalling is only highly activated on these undifferentiated cells in a non-autonomous manner. In addition, the authors found that the ratio of transcript level of spitz-to-argos is determinant for proper activation of the pathway and for the establishment of the ommatidial pattern.

In general, the study provides some interesting observations, especially the non-autonomous effect of spitz and argos, and contributes to a better understanding of eye development and the modulation of the activity of EGFR signaling pathway. However, I have some important concerns regarding the results of this study.

My main concern is that the ratio of spitz/argos regulation of EGFR signalling pathway is mainly based on the fact that depletion of spitz in the PR cells using elavGal4 fails to induce any phenotype in the adult eye. Under these conditions, the levels of spitz are only reduced, rather than abolished, probably due to the strength of either the driver or the dsRNA. Consequently, the activation of argos is proportionally diminished and a mild effect on the determination of PC was observed, as shown in figure 4 where Elav expression and morphogenetic furrow proliferation are slightly affected. In contrast, when spitz was knock down with a more general eye driver, GMR, expression of both spitz and argos are virtually eliminated and EGFR signaling is not activated as Elav expression and morphogenetic furrow proliferation are almost abolished (Fig. 4). In this line, what is the level of dpERK in elav> spitzRNAi compare to GMR>spitzRNAi?

Based on these observations, the authors claim that the phenotype of the eye adult fly is due to the loss of the proper spitz-to-argos expression ratio rather than a mild reduction of EGFR signaling in the first case and almost complete inactivation of the pathway.

Since argos expression depends on EGFR signaling activation by spitz, the only factor that is critical to modulate EGFR is the level of spitz. As shown by the authors, reduction of spitz mRNA in PR cells induces a concomitant reduction of argos expression in the NP cells, meaning that spitz:argos ratio only depends on the expression of spitz.

Similarly, the authors reach the same conclusion when argos was depleted in the GMR domain. According to the results shown on figure 2, dsRNA of argos seem to work better as its mRNA transcripts were effectively abolished, as a consequence the ration of spitz/argos is modified and therefore EGFR signaling is wrongly activated. On other words if you eliminate completely argos or spitz is obvious that the ommatidial pattern should be affected and consequently not conclusive.

In the same line, the experiments to modified the ratio using EGFR mutants are also not conclusive. Both EGFRCA and EGFDN induced additional effects on the developing eye (both induces less proliferation in the MF and opposite effects on Elav expression, Figure 4). For instance, depletion of spitz and overexpression of EGFRDN produce opposite effects on elav expression. Similarly, overexpression of argos and EGFRDN also produce different eye phenotypes, making difficult to interpret the results in terms of expression ratios.

A better approach to investigate the ratio spitz-argos might be to play directly with the ligands level in the cells where they are expressed. Thus, the authors have already shown that, as expected, reducing the levels of spitz in elav positive cells proportionally reduced the expression of argos (Figure 2). However, what happens when spitz is overexpressed? Does the, overexpression of spitz using elavgal4 increase argos expression in NP in the same proportion than in the wt? On the other hand, the authors modified argos levels by overexpressing it in all eye cells. Under these conditions, argos still acts in a non-autonomously manner? How sure the authors are, that the phenotype observed is not produces by and ectopic expression of argos in cells that usually does not produce this protein rather than a change in the expression ratio? Is it possible to increase argos levels specifically in NPR cells? What is the effect of overexpressed of argos in PR cells, using elavGal4?

Finally, I do not understand why the authors claim that the ratio spitz/argos is around 2,1. If you compared the expression of spitz in PR cells vs. argos in the NP cells the mRNA/1000nm3 are nearly the same. Therefore, the ratio is around 1:1 as previously reported.

Other comments that might help the authors to improve the Ms.

1) Addition of the expression domains of ElavGal4 and GMRgal4 might help to non-specialist readers to better understand the logic of the experiments.

2) In Figure 1 the authors show the complementary expression of spitz and argos in Elav positive cells and non-expressing Elav cells. However, whereas the non-overlapping expression of the two ligands is quite clear, the fact that are elav positive cells or not is not properly shown as in the merge panel is difficult to distinguish the blue color.

3) In figure 2, depletion of argos under the control of GMRGal4 induce a significance increase in the expression of spitz. However, overexpression of neither EGFRDN nor EGFRCA do not change the expression of spitz, suggesting a different effect on signalling when using these constructs.

4) Significance of the differences in mRNA counts should be added in the graphs of all figures.

5) Since there are argos and spitz fusion proteins or Ab. Could and extracellular staining of the proteins use as a tool to determine the ratio spitz/argos?

6) The changes in the expression of spitz and argos is shown in the overexpression of EGFRCA but not in the UASargos or UASEGFRDN.

7) Adult heads are not properly orientated anterior-posterior in Figure 3F, 4F and SFig 2B.

**Have all data underlying the figures and results presented in the manuscript been provided?**

Reviewer #1: None

Reviewer #2: Yes

Reviewer #3: Yes

PLOS authors have the option to publish the peer review history of their article (what does this mean?). If published, this will include your full peer review and any attached files.

Reviewer #1: No

Reviewer #2: No

Reviewer #3: No

---

## [Editor Report · Decision Letter 1]

17 Jan 2023

Dear Dr Mazumder,

I read your revised manuscript as well as your response to reviewer comments and I am pleased to inform you that your manuscript entitled "Buffered EGFR signaling regulated by spitz-to-argos expression ratio is a critical factor for patterning the Drosophila eye" has been editorially accepted for publication in PLOS Genetics. Congratulations! I want to thank you for the effort that you put into the revision and the thoughtfulness of your response to suggestions. I feel that you manuscript is greatly improved and will make an important contribution to our understanding of both Drosophila eye development and EGFR signaling. 

Yours sincerely,

Justin Kumar, PhD

Guest Editor

PLOS Genetics

Gregory P. Copenhaver

Editor-in-Chief

PLOS Genetics

Comments from the reviewers (if applicable):

**Data Deposition**

http://datadryad.org/submit?journalID=pgenetics&manu=PGENETICS-D-22-01063R1

**Press Queries**

---

## [Editor Report · Acceptance letter]

30 Jan 2023

PGENETICS-D-22-01063R1 

Buffered EGFR signaling regulated by spitz-to-argos expression ratio is a critical factor for patterning the Drosophila eye 

Dear Dr Mazumder, 

We are pleased to inform you that your manuscript entitled "Buffered EGFR signaling regulated by spitz-to-argos expression ratio is a critical factor for patterning the Drosophila eye" has been formally accepted for publication in PLOS Genetics! Your manuscript is now with our production department and you will be notified of the publication date in due course.

With kind regards,

Timea Kemeri-Szekernyes

PLOS Genetics

On behalf of:
